# Molecular Dynamic Simulations for Biopolymers with Biomedical Applications

**DOI:** 10.3390/polym16131864

**Published:** 2024-06-29

**Authors:** Ramón Garduño-Juárez, David O. Tovar-Anaya, Jose Manuel Perez-Aguilar, Luis Fernando Lozano-Aguirre Beltran, Rafael A. Zubillaga, Marco Antonio Alvarez-Perez, Eduardo Villarreal-Ramirez

**Affiliations:** 1Instituto de Ciencias Físicas, Universidad Nacional Autónoma de México, Cuernavaca 62210, Mexico; ramon@icf.unam.mx; 2Laboratorio de Bioingeniería de Tejidos, División de Estudios de Posgrado e Investigación, Coyoacán 04510, Mexicomarcoalv@unam.mx (M.A.A.-P.); 3School of Chemical Sciences, Meritorious Autonomous University of Puebla (BUAP), University City, Puebla 72570, Mexico; 4Centro de Ciencias Genómicas, Universidad Nacional Autónoma de Mexico, Cuernavaca 62210, Mexico; 5Departamento de Química, Universidad Autónoma Metropolitana-Iztapalapa, Mexico City 09340, Mexico; zlra@xanum.uam.mx

**Keywords:** biopolymers, molecular dynamics simulations, polymeric materials, biopolymer, protein absorption

## Abstract

Computational modeling (CM) is a versatile scientific methodology used to examine the properties and behavior of complex systems, such as polymeric materials for biomedical bioengineering. CM has emerged as a primary tool for predicting, setting up, and interpreting experimental results. Integrating in silico and in vitro experiments accelerates scientific advancements, yielding quicker results at a reduced cost. While CM is a mature discipline, its use in biomedical engineering for biopolymer materials has only recently gained prominence. In biopolymer biomedical engineering, CM focuses on three key research areas: (A) Computer-aided design (CAD/CAM) utilizes specialized software to design and model biopolymers for various biomedical applications. This technology allows researchers to create precise three-dimensional models of biopolymers, taking into account their chemical, structural, and functional properties. These models can be used to enhance the structure of biopolymers and improve their effectiveness in specific medical applications. (B) Finite element analysis, a computational technique used to analyze and solve problems in engineering and physics. This approach divides the physical domain into small finite elements with simple geometric shapes. This computational technique enables the study and understanding of the mechanical and structural behavior of biopolymers in biomedical environments. (C) Molecular dynamics (MD) simulations involve using advanced computational techniques to study the behavior of biopolymers at the molecular and atomic levels. These simulations are fundamental for better understanding biological processes at the molecular level. Studying the wide-ranging uses of MD simulations in biopolymers involves examining the structural, functional, and evolutionary aspects of biomolecular systems over time. MD simulations solve Newton’s equations of motion for all-atom systems, producing spatial trajectories for each atom. This provides valuable insights into properties such as water absorption on biopolymer surfaces and interactions with solid surfaces, which are crucial for assessing biomaterials. This review provides a comprehensive overview of the various applications of MD simulations in biopolymers. Additionally, it highlights the flexibility, robustness, and synergistic relationship between in silico and experimental techniques.

## 1. Introduction

Biopolymers are vital in the overlap between materials science and biomedical engineering, encompassing naturally occurring substances derived from living organisms and engineered synthetic materials [1,2]. These materials are precisely engineered with the ultimate goal of serving as replacements or aids in repairing tissues and organs [3,4]. Biopolymers, including lipids, polysaccharides, proteins, and nucleic acids, are synthesized by bacteria, fungi, plants, and animals through millions of years of evolution. These natural polymers exhibit a wide range of specific functions, often surpassing the properties of man-made materials [5,6]. An essential requirement for biomaterials is their ability to interact with cells, facilitating their intended function while eliciting an appropriate host response in a given application [4]. However, cells colonizing the surfaces of biomaterials do not come into direct contact with them; instead, cells recruit serum proteins or secreted proteins that adhere to the surfaces. Consequently, protein adsorption on biomaterial surfaces plays a key role in determining biocompatibility [7]. 

Most studies investigating the interactions between proteins and material surfaces focus on the kinetics of protein adsorption at the macromolecular level [8]. These studies often overlook the underlying forces governing protein adsorption on diverse biomaterial surfaces and the structural changes that proteins may undergo upon adhesion to these surfaces [9,10]. The applied techniques include X-ray photoelectron spectroscopy (XPS), static secondary ion mass spectrometry (SSIMS), scanning tunneling microscopy (STM), surface plasmon resonance (SPR), and solid-state nuclear magnetic resonance (NMR), which offer the ability to obtain quantitative information about protein–surface interactions at different levels of detail [9,11]. Several studies have demonstrated that factors such as surface structure, hydrophobicity, and charge profoundly influence the structural characteristics of proteins upon their adsorption to solid–liquid interfaces. The release of water is a critical step in the adsorption of proteins onto biomaterial surfaces. This process forms two distinct interfaces: the surface–water interface and the protein–water interface. The adsorption of proteins onto the surface can induce structural alterations, potentially leading to protein denaturation [12]. These structural changes upon protein adsorption are due to the native folding corresponding to the free energy minimum in solution and do not correspond to the free energy minimum once the protein is in contact with the surface [13]. To comprehensively understand protein adsorption phenomena at the molecular level, we must consider the protein–surface interface as a system that contains at least three components: a surface, a protein, and a solvent. In aqueous systems, the structural arrangement of water molecules near the surface and the solutes induces water-mediated interactions between them. These interactions play a significant role in the direct interactions between the surface and the solute, exhibiting attractive (e.g., hydrophobic) or repulsive forces depending on the chemistry and nature of the surface and the solute. When dealing with flexible macromolecular conformations, additional relevant questions arise regarding the preferred conformations in the adsorbed state and the dynamics of conformational transitions at the interface [14].

Water plays a crucial role in biological processes and significantly influences the structure and function of biomolecules. 

The properties of water, particularly the dynamics of the hydrogen bond network and hydrophobicity, have been extensively studied to elucidate the interaction between protein surfaces and the surrounding layers of water molecules [15]. Hydrophobicity is a crucial property of biopolymers, as it determines their interactions with water molecules [16]. Hydrogen bonding is a critical factor in determining the solubility and stability of biopolymers in water. MD simulations can also determine the ability of biopolymers to form hydrogen bonds with water molecules [16]. These surrounding layers form a hydration shell, which exhibits distinct properties compared to the bulk water in distant layers [17]. The topography and chemical composition of the protein surface interact with and alter the orientation of individual water molecules, leading to dynamic differences between the first few layers of water and the more distant layers that resemble bulk water properties [18]. The biomolecular surface dictates the conditions under which water molecules in the hydration shell exhibit variations in their rotational and vibrational degrees of freedom compared to water molecules farther away from the protein surface [19].

## 2. Basics of Molecular Dynamics Simulations

Alder introduced MD in 1959 [20]. This method aims to solve Newton’s equations of motion for a defined set of molecules within a virtual system [20]. MD simulation is a powerful method for investigating the time evolution of a set of molecules or atoms within a virtual environment. It allows for molecules to be evaluated within in silico systems for several properties [21]. Through MD simulations, one can assess the molecular conformations and energies of the system at finite temperatures, explore the nature of structural and energy transition states, and obtain a dynamic atomic landscape that is otherwise inaccessible through experimental techniques, among many other properties [22]. The success of this method stems from two key factors. Firstly, virtual systems can be tested and compared directly with experimental results, providing valuable insights that are complementary to those obtained via experimental characterization. Secondly, MD simulations offer a glimpse into processes that may not be directly observable in experiments. However, it is essential to acknowledge the limitations of these methods, such as the finite number of particles, the computational processing capacity required, and the inherent simplifications made when assuming the behavior of molecules in virtual systems mirrors that of real systems [22].

The cornerstone of life is contingent upon the intricate and perpetually evolving interplay of complex chemical networks at the molecular level. This encompasses myriad processes, including biopolymer-mediated phenomena such as protein folding, nucleic acid dynamics, ion transport across membranes, and the enzymatic catalysis of biochemical reactions [23]. Given the inherent complexity and dynamic nature of biological systems, computational simulation methodologies have emerged as crucial tools, especially with the advancement of powerful in silico workstations [24]. While many modeling techniques traditionally focus on static molecules, for instance, performing molecular mechanics (MM) calculations at 0K, which essentially freezes the molecule, there is a limitation in capturing the natural atomic motion inherent in these studies. Therefore, there is a compelling need for theoretical simulation systems that can provide dynamic insights into the behavior of chemicals and biomolecules, offering a valuable complement to experimental investigations [25].

MD can be defined as a computer simulation technique that permits predicting the time evolution of a particular system, involving the generation of atomic trajectories using the numerical integration of Newton’s equation of motion for a specific interatomic potential defined by an initial condition and boundary condition [20]. The dynamic simulation also provides information on molecular kinetics and thermodynamics [26]. Determining the time-dependent motion of individual particles of a system allows for quantifying the properties of the given system on a definite time scale that is otherwise unattainable [26]. Most importantly, MD simulation development can be attributed to the revolutionary advancements of computational algorithms and technology, which allowed MD to be applied in several areas of chemistry and physics [27]. In particular, determining the time-evolution position of atoms allowed us to identify specific conformational variations, for instance, changes in the secondary structural content of a protein domain that may be relevant to the protein’s function (or dysfunction). This was the case of the Huntingtin exon 1 domain, where all-atom MD simulations revealed a change in the secondary structural propensity of the polyglutamine segments, from β-strands to α-helices, based on the interaction with polyproline contiguous segments [28]. Furthermore, the changes in a particle’s position information under the conditions of the simulations can provide dynamic knowledge about the interaction of ions with particular amino acid residues, enabling us to define allosteric binding sites. This was the case with the discovery of a new sodium-binding site located on the extracellular side of the cannabinoid type 1 receptor [29]. Figure 1 shows the simulated molecular system with the location of the sodium binding site. Lastly, in the context of Alzheimer’s disease, unbiased MD simulations were utilized to understand the difference in the protonation state of the β-amyloid peptide in favoring the stability of different protofibril conformations [30]. The researchers found that the deprotonated state stabilized the protofibril by forming a charge–charge interaction with a basic residue in the same peptide, which provided insight into the polymorphism observed in Aβ fibrils. 

MD simulations are applied in two major approaches to address a physical system, considering the nature of the model and the mathematical formalism involved. These approaches include classical mechanics and quantum chemical formalisms, as described below:The classical mechanics approach considers molecules as classical objects similar to the ball-and-stick model, where atoms represent soft balls and bonds represent elastic sticks. The dynamics of a system are determined by the laws of classical mechanics in this approach [31].The quantum mechanics approach, also known as first-principles MD simulation, was first introduced by Car and Parrinello in 1985 [32]. This approach considers the quantum nature of chemical bonds. In this method, the bonding within a system is determined by applying quantum equations to the electron density function of the valence electrons, while the dynamics of ions (nuclei with their inner electrons) are treated classically. Quantum MD simulations are an improvement over classical simulations, and they provide valuable information on various biological problems. However, they consume more computational resources than classical simulations [31].

The core requirement for MD simulation is quite simple; it involves a set of conditions defining the initial positions and velocities of all particles and the interaction potential that defines the forces among all the particles. Second, the determination of the evolution of the system in time is achieved by solving a set of equations of motion for all particles considered in the system. In the case of classical mechanics, Newton’s law is applied to define the motion of classical particles. Even a classical MD simulation for biomolecular systems consisting of thousands of atoms over a nanosecond time scale consumes significant computational resources [33]. See Figure 2 for the computational workflow of MD simulations.

The MD simulation formalism may comprise five conditions: boundary condition, initial condition, force calculation, integrator/ensemble, and property calculation. The classical MD simulations boil down to numerically integrating Newton’s equations of motion for the particles (atoms, in the simplest case) that build up the investigated system:(1a)$$m\frac{d^{2}r_{i}(t)}{{dt}^{2}}=F_{i}\left(r_{1},r_{2},\cdots ,r_{N}\right),\;\;\;\;\;\;\;\;\;\;i=1,\;2,\;\cdots ,\;N$$

Here, ***r_i_*** are the position vectors and ***F_i_*** are the forces acting upon the *N* particles in the system. 

Quite often, forces are derived from potential functions, ***U*** (***r***_1_, ***r***_2_, …, ***r_N_***), representing the potential energy of the system for the specific geometric arrangement of the particles:$$F_{i}\left(r_{1},r_{2},\cdots ,r_{N}\right)=-\nabla_{r_{i}}U\left(r_{1},r_{2},\cdots ,r_{N}\right)$$

This form implies the conservation of the total energy *E = E_kin_ + U*, where E_kin_ is the instantaneous kinetic energy.

In the absence of external forces, the potential can be represented in the simplest case as a sum of pairwise interactions:(1b)$$U=\sum_{i=1}^{N}\sum_{j>i}^{N}u\left(r_{ij}\right)$$where ***r_ij_*** = ***r_i_*** − ***r_j_***, ***r_ij_*** ≡ |***r_ij_***| and the condition ***j*** > ***i*** prevent the double counting of particle pairs. The forces acting on the particles are composed in such a case of the individual interactions with the rest of the particles:(1c)$$F_{i}=\sum_{j\neq i}^{N}f_{ij},\;\;\;\;\;\;\;f_{ij}=-\frac{du\left(r_{ij}\right)}{dr_{ij}}\cdot \frac{r_{ij}}{r_{ij}}$$

According to Newton’s third law, ***f_ji_*** = −***f_ij_***. The computational effort needed to solve the set of equations of motion (1a) is proportional to N^2^ and is mainly associated with the evaluation of forces. Therefore, for tractable computations, the forces should be expressed analytically. To further reduce the computational effort, the potential can be cut off at some limiting separation (for ***r****_ij_* > ***r****_cut_*) beyond which the potential becomes negligible.

Equation (1a) is a differential equation of the second order that can be integrated by providing specific values of the initial positions of particles, their velocities, and the instantaneous force acting on them. The equation of motion is discretized, followed by a numerical solution because of the many-body system comprised of the particles. The trajectories in MD simulation are defined by position and velocity vector components, and the time evolution of the system is depicted in phase space. The position and velocity components are promulgated with a finite time interval by employing numerical integrators. The position of each particle in space is designated by ***r****_i_*(*t*), while the kinetic energy and temperature of the system are determined by velocity ***v****_i_*(*t*). Figure 3 shows the vector components of a particle trajectory. The position and velocity components are promulgated with a finite time interval by employing numerical integrators. The specialty of MD simulation is that it allows a direct tracing of the dynamic events that might be influential to the functional properties of the system.

The integration of Newton’s force equation is performed to obtain an expression that gives the position ***r_i_***(*t* + Δ*t*) at time *t* + Δ*t* in terms of the already-known positions at time *t*. By employing the Taylor series, the mentioned position can be mathematically represented as follows:(2)$$r_{i}\left(t+\Delta t\right)≅2r_{i}\left(t\right)-r_{i}\left(t-∆t\right)+\frac{F_{i}\left(t\right)}{m_{i}}∆t^{2}$$

The time step (Δ*t*) for most MD simulations is on the femtosecond scale, which is the scale of chemical bond vibrations. MD simulations are limited by the highest-frequency vibration and the time step should be ten times lower than the highest frequency.

Velocity calculations in MD programs can be performed using either position-based or explicit methods, such as the leapfrog and velocity Verlet schemes [34,35]. Trajectory files typically store positions and velocities with infinitesimally small integration steps. However, longer time steps are necessary for sampling longer trajectories. In coarse-grained simulations, larger-mass atoms are used, which require longer integration times and result in longer trajectory lengths [36]. Another crucial aspect of MD simulation is its statistical mechanics-based behavioral nature, enabling the averaging of values obtained at a microscopic level. The Newtonian dynamics adhere to the conservation of energy, and MD trajectories provide a microcanonical ensemble distribution of configurations. MD simulation is a technique that allows the measurement of physical quantities by averaging instantaneous values from trajectories. It can simulate a variety of experimental conditions, such as protein in a vacuum, crystal environment, and explicit water environment [37]. Unfortunately, molecular dynamics simulations have limitations such as their finite system size, limited observation time, and inability to form or break covalent bonds [32].

## 3. Interactions of Waters with Biopolymers

When considering biopolymer interactions with water molecules, we lack detailed molecular-level knowledge about the structures, dynamics, hydrogen bonds, and van der Waals networks [38]. There are two approaches to analyzing the interface interaction between water and biopolymers. Some studies analyze the water dynamics through simulations and experiments, to understand the effect of the biomolecule on the surrounding water molecule layers or hydration shell. Other studies focus on how water may determine the structure and dynamics of biopolymers [15,18,19].

Several experimental methods—a solution-state and solid-state NMR spectroscopy [20,21], X-ray diffraction [39], neutron scattering [40], and fluorescence spectroscopy [41]—have been used to analyze the interactions between water and biopolymers. These methods measure the rotational and translational dynamics of water molecules and can detect different physical phenomena. Recently, terahertz (THz) spectroscopy has been utilized to characterize molecular structures and the hydration of biopolymers, enabling the analysis of the dynamic hydrogen bond network during biochemical reactions [42,43]. THz spectroscopy has been helpful in determining the effect of biopolymer surfaces on water molecules, revealing that biopolymers can influence water even several layers away [44].

MD simulations have been used in conjunction with experimental observations to gain a deeper understanding of the interaction between the hydration shell and the biopolymer surface characteristics [45,46,47]. However, while we know much about the structure and dynamics of bulk water, the behavior of water molecules at the hydration shell is not as well understood. The interface between the protein surface and the surrounding water is notably diverse, making molecular simulations more challenging. Amino acid lateral chains, possessing charged and polar groups, engage in hydrogen bonds and electric fields with adjacent water layers, thereby altering the rotational and vibrational degrees of freedom of water molecules. Additionally, nonpolar atoms enhance interactions between neighboring water molecules [44]. These diverse protein properties collectively influence the interaction of the first hydration layers with water, extending even to those distant from the protein surface. Serratos et al. [48] conducted research demonstrating trapped water molecules at the interface using both in silico and experimental techniques. They used fluorimetry and Multithermal Titration Calorimetry experiments to show that the affinity of phosphorylated inhibitors 2PG and PGH for triosephosphate isomerase gradually decreases with the addition of osmolytes. The less negative binding heat capacity observed in these experiments is indicative of water molecule entrapment. MD simulations in explicit solvent with the free enzyme revealed a fixed presence of only 3 water molecules at the binding site for at least 5 ns. In contrast, complexes with PG and PGH retained 10 and 8 water molecules, respectively. Thus, the binding of these charged ligands leads to the entrapment of water molecules at the site [48].

In order to model the hydration of biopolymers, it is necessary to have prior knowledge of polymer structures and to develop specialized water models. Traditionally, force field approaches have been used to accurately simulate protein folding, protein dynamics, and ligand–receptor binding. Force fields provide information about the potential energy of particles in a system, including both bonded interactions (such as bond lengths, bond angles, and torsional dihedral angles) and non-bonded interactions (such as Van der Waals forces and electrostatic forces) [45]. When simulating the interaction between proteins and their surrounding hydration shell, protein force fields are used in conjunction with specific water models that are incorporated into software packages like CHARMM, AMBER, and GROMOS, which are among the most widely used first-generation force fields [49]. A list of programs available for MD simulations and their corresponding links can be found in Table 1.

MD simulations frequently serve as a complementary approach to elucidate molecular-level behaviors at interfaces [50]. MD simulations simplify systems by representing a thin polymer/air/water interface, enabling a direct analysis of the distribution of all interface components, surface pressure profiles, hydrogen bonding patterns, impacts of oil droplets, and alterations in orientation [51].

The relationship between water and polymers, especially in contexts such as electronics, remains a persistent subject of concern. The ingress of water can significantly impact the physical properties of polymers and the integrity of composite structures [38]. Molecular dynamics simulations at the molecular level provide detailed insights into how water interacts with polymers and interfaces, allowing for the identification of areas most susceptible to water-related effects and facilitating an understanding of changes in adhesion and integrity. These simulations enable observations of changes in properties such as surface tension, interface thickness, and water molecule diffusion in polymeric systems [38].

Consider a study involving the gradual addition of water to a copolymer, revealing a decrease in the radius of gyration of the polymer chain and an increase in cohesive energy density. Notably, the diffusion coefficient initially decreases and subsequently increases with increasing water concentration, a phenomenon mirrored by the fractional free volume. These observations shed light on how water can modify both the physical properties and the molecular structure of the polymer [51]. In the interfaces involving surfactants and polymers, molecular dynamics simulations elucidate the arrangement of surfactants within the liquid film. At low surface concentrations, surfactants form small clusters at the interface, whereas intermediate or high concentrations entirely coat the interface. Furthermore, the orientation of surfactant tails changes with increasing surface concentration. Additionally, the role of polymers in retaining water and influencing the mobility of water molecules at the interface through hydrogen bonding is emphasized [50].

These molecular modeling approaches collectively provide a more profound understanding of how water interacts at the molecular level with polymers and interfacial surfaces. Such insights are vital for enhancing the reliability and properties of a wide range of applications, spanning from electronics to the stabilization of wet foams in the oil, chemical, and food industries [38].

In the dynamic field of research into polymers and their interactions with water, molecular and mesoscale simulations have become indispensable tools, proving effective in applications from novel polymer development to understanding mechanisms in electronics and beyond [38]. A key concept emerging from these simulations is the pivotal importance of interfaces. In the context of polymer–water interactions, these interfacial regions profoundly shape material properties and behavior. Molecular simulations define and characterize these interfaces, and so they are crucial for guiding experimental research and materials’ development [38].

One intriguing facet involves exploring how water affects polymer properties at the molecular level. Simulations unveil its impact on various characteristics, including molecule diffusion, cohesion, adhesion, and deformation. Furthermore, they identify the underlying molecular structures contributing to these behavioral alterations of materials [38].

Notably, molecular dynamics simulations reveal diverse responses of different polymers to water, with the polarity and molecular structure playing pivotal roles. This underscores the unique capacity of molecular simulations to pinpoint specific molecular species or structures responsible for distinct physical responses [38].

Integrating mesoscale simulations with molecular models has provided a more comprehensive perspective, particularly in addressing surface roughness [38]. These integrated approaches not only shed light on the morphological effects of water at interfaces but also elucidate its impact on material mechanics [50].

## 4. Interactions of Nanostructures with Biopolymers

With the recent advancement of nanoscience and nanotechnology, particularly in the areas of biomedical research, the biocompatibility of these materials, that is, the way they interact and their behaviors in the context of biological systems, has become a central issue [52]. Solid–water interfaces are ubiquitous in biological, colloidal, and soft condensed matter systems, where proteins, biopolymers, surfactants, polymers, colloids, and other solutes often exhibit interfacial activity and adsorb to various interfaces [53]. Unwanted adsorption poses a significant challenge in applications ranging from separation processes to implant wettability and bioactivity. Conversely, there is a need for precise control and engineering of binding for specific purposes, such as molecule alignment or pattern creation [54]. These findings have practical implications in sensing, detection, facilitating precise coordination with water molecules [53], and orchestrating the regulation of protein adsorption [54].

Due to their unique volume-to-surface ratios, nanomaterials exhibit novel properties. The diverse sizes, shapes, and surface chemistries of nanoparticles can significantly influence their toxicity, gene-regulating effects, and clearance within biological systems [55]. These effects arise from the intricate interactions between nanoparticles and biological molecules at the atomic level [55]. The connection between nanostructures and their in vivo biodistribution remains uncertain, mirroring our incomplete comprehension of the interactions between nanomaterials and biological tissues [55].

The physicochemical mechanisms that nanoparticles undergo to affect biological molecules and complex systems are currently under investigation by experimental and theoretical methods. However, the study of nanoparticle interactions with biological environments at the nanoscale is limited by the current experimental techniques. Concurrently, the spatial and temporal resolutions enabled by computational techniques allow for examining specific interactions and dynamics induced by nanoparticles in biological molecules. These advancements have led to the increasing use of computational methods to elucidate the fundamental aspects of intermolecular interactions within nanoparticle–biomaterial systems [39].

Here, we provide a summary of different computational studies that have shed light on the role of the nanostructure in biological environments. From the myriad nanostructures and surfaces important for the interaction with biopolymers, we will concentrate our discussion on the interactions of proteins with carbon-based and with noble-metal-based materials since they comprise the most promising materials for biomedical applications. Lastly, we will discuss the interaction with relevant biominerals, with particular attention to their influence on the conformational behavior of intrinsically disordered proteins.

### 4.1. Interaction of Carbon-Based Nanomaterials with Proteins

Among various nanostructures, carbon-based nanomaterials, such as fullerenes, graphene, and carbon nanotubes, are extensively studied due to their exceptional mechanical and electrical properties.

#### 4.1.1. Fullerenes and Fullerene Derivatives

Fullerenes are closed cage structures made of carbon atoms with sp^2^ hybridization that can be rapidly functionalized to modify their physicochemical properties for specific applications. Computational approaches have characterized the interaction between fullerenes (e.g., C60) and the human immunodeficiency virus 1 (HIVP) protein, a promising target for antiviral agents [56]. Considering the protein’s catalytic site, the C60 molecule was deemed a potential inhibitor of HIVP [56]. The protein complex was modeled and docked with a C60 carbon-based nanostructure to identify specific interactions [57]. The results suggest that the C60 structure adequately fits into the catalytic site and strongly interacts with the protein’s hydrophobic regions, inhibiting its enzymatic functions [57].

A water-soluble C60 derivative was also investigated using similar computational approaches, yielding comparable results [57]. Computational findings were validated through experimental assays using the water-soluble C60 variant. Finally, the inhibitory properties of the C60 derivative were examined in cellular-based assays, demonstrating significant and selective toxicity toward viral-infected cells [57].

Experimental evidence has demonstrated the capacity of some proteins to specifically recognize C60 derivative molecules, particularly some murine antibodies [58,59]. Here, computational techniques were used to identify the particular interactions involved in the formation of this protein–nanostructure complex [60]. Docking calculations suggested particular complex structures in the variable regions investigated by molecular dynamics simulations. The computational results support the suggested molecular poses and show a significant rotational motion by the nanostructure [60]. Also, the results suggested that the high affinity and specificity are due to structural complementarity and significant interactions involving protein side chains (e.g., aromatic interactions).

The capacity of specific protein recognition of C60 derivative molecules, particularly certain murine antibodies, has been demonstrated through experimental evidence [58,59]. In this context, computational techniques were employed to identify the specific interactions involved in the formation of complexes between proteins and nanostructures [60]. These complex structures in the variable regions were suggested by docking calculations, and they were subsequently investigated through molecular dynamics simulations. The computational results not only corroborate the proposed molecular orientations but also reveal significant rotational motion exhibited by the nanostructure [60]. Furthermore, the results imply that the observed high affinity and specificity are due to structural complementarity and significant interactions, notably involving protein side chains such as aromatic interactions [60]. This particular result indicates a possible widespread role of this aromatic interaction (π-π stacking) in the interaction between the carbon-based nanostructure and proteins [60].

Hydroxylated fullerenes encapsulating a metal ion, known as metallofullerenols, have undergone extensive experimental and computational characterization. Specifically, [Gd@C_82_(OH)_22_]*_n_* has been shown to inhibit the catalytic function of matrix metalloproteinase-2 and -9 (MMP-2 and MMP-9) [61]. These proteins play crucial roles in angiogenesis, the formation of new blood vessels, and extracellular matrix degradation, making them pivotal in cancer metastasis. Computational methods, particularly molecular dynamics simulations, complement experimental assessments of these nanoparticles. MD simulations have revealed the precise binding site of [Gd@C_82_(OH)_22_]*_n_* within the MMP-9 structure, selectively suppressing its catalytic activity and providing atomic-level insights into the tumor-suppressing attributes of these nanoparticles [61]. Remarkably, in contrast to carbon-only molecules, [Gd@C_82_(OH)_22_]*_n_* nanoparticles cause minimal disruptions to the 3D structure of proteins [61].

Beyond their anticancer activities, [Gd@C_82_(OH)_22_]*_n_* nanoparticles have been observed to inhibit protein–protein interactions. Two systems illustrating this property involve direct and indirect interactions of these nanoparticles with the WW- and SH3-domains, respectively. Through computational simulations, a direct interaction between [Gd@C_82_(OH)_22_]*_n_* nanoparticles and the WW domain was identified, particularly with the native ligand binding site that recognizes proline-rich motifs [62]. Additionally, computational approaches indicated a higher binding affinity of the WW domain for [Gd@C_82_(OH)_22_]*_n_* nanoparticles compared to a proline-rich peptide [62]. Regarding the SH3 domain, simulation results suggest that, even though [Gd@C_82_(OH)_22_]*_n_* nanoparticles are capable of interacting with the native binding site of the SH3 domain, the primary inhibitory effects result from the interaction with a region of the protein responsible for recognizing and guiding the proline-rich motif ligand toward the native binding site [63].

#### 4.1.2. Carbon Nanotubes

Carbon nanotubes (CNTs) are cylindrical nanostructures composed of sp^2^ carbon atoms, which have attracted significant attention for their outstanding electrical and mechanical properties, which make them an attractive material for many industrial and biomedical applications [64].

Interactions between CNTs and proteins have been extensively documented, from interactions of lung proteins to proteins present in the circulatory system [65]. To date, our knowledge of the specific mechanisms of interaction at the molecular level has primarily arisen from computational studies [66]. Two main mechanisms have been proposed for functional modulation by CNTs: disruption of protein active sites and competition with native ligands.

The interaction of CNTs and lung proteins has been comprehensively studied, especially concerning potential adverse consequences, and has yielded valuable insights. One of these studies involved the interaction of CNTs with members of the collenctins protein family, specifically the surfactant proteins SP-A and SP-D, integral components of the immune pulmonary defense system [67]. The outcomes reveal a discernible preference for interaction between these proteins and CNTs, which suggests a possible molecular explanation for the toxicity of these nanomaterials. This finding also suggests that modifying the CNTs could reduce their harmful effects, but this approach requires careful consideration [68].

Similarly, the interaction of CNTs and proteins in the circulatory system has provided details about the possible effect of the binding. In a detailed investigation, the interaction of single-walled CNTs and four common plasma proteins was studied using different techniques, including computer simulations [69]. The results present a model where proteins compete to bind the nanomaterial. Interestingly, the results indicate that upon absorption, the integrity of secondary structural elements of the biomolecules becomes compromised and that the degree of the disruption is related to the binding strength [69].

Due to their capability for scrutinizing molecular-level interactions, computer simulations have illuminated the intricacies of CNT interactions with biopolymers. MD simulations have given rise to two predominant hypotheses regarding the roles of CNTs: one involving the perturbation of protein active sites and the other entailing competition for protein binding sites with native ligands.

Insights into the first hypothesis were gleaned from simulations of the interaction between single-walled carbon nanotubes (SWCNTs) and the WW domain [70]. Computational findings suggest that SWCNTs exhibit a strong affinity for the WW-domain hydrophobic core, thereby disrupting the structure of the native binding site [70]. Furthermore, these results indicate the absorption of the native ligand of the WW domain, a proline-rich motif, onto the nanomaterial. Through this mechanism of disruption and obstruction, SWCNTs incapacitate normal function proteins [70].

Evidence supporting the second hypothesis arises from the ability of carbon nanotubes to outcompete native ligands for receptor binding. Remarkably, SWCNTs exhibit a stronger binding affinity to SH3 domains than native proline-rich ligands while preserving the protein tertiary structure [71]. In this scenario, the nanomaterial binds to the same receptor pocket as the native ligand through robust hydrophobic and aromatic interactions, thereby impeding the protein standard recognition function [71].

#### 4.1.3. Graphene and Graphene Derivatives

Graphene, a single-sheet nanostructure comprising sp^2^ carbon atoms arranged in a hexagonal lattice, shares resemblances with other carbon-based nanomaterials and finds diverse applications, notably in biomedical devices [72]. However, its interaction with biomolecules has been studied less than other carbon-based materials. To address this gap, MD simulations were conducted to investigate the interactions between the miniprotein Villin headpiece and graphene [73]. Their findings unveil a comprehensive mechanism for protein adsorption driven by the initial interaction of an aromatic residue, F35, located at the C-terminal. This interaction aligns its side chain parallel to the graphene sheet. Subsequent interactions involve multiple aromatic residues. Upon adsorption onto the nanomaterial, the protein undergoes a significant disruption of its helical secondary structure due to several aromatic residues adopting parallel side chain conformations, resulting in a loss of structural integrity [73]. During these simulations, several aromatic residues placed themselves in parallel side chain conformations relative to the nanomaterial, which caused a loss of structural integrity [73]. Not only have the interactions of prototypical proteins and pristine graphene been investigated but also a variety of graphene derivatives including several carbon nitrite nanomaterials. For instance, it was reported that nitrogenated graphene (C_2_N) displayed mild interactions with a prototypical protein causing no disruption in the protein’s structure [74]. Moreover, both the nanosheets C_3_N_3_ and C_3_N_4_ exhibit adequate biocompatibility in the interaction with model globular proteins, that is, the tertiary structure of the proteins remains stable upon absorption on the 2D nanomaterial [75,76]. In contrast, all-atom MD simulations revealed that upon adsorption, the carbon nitride polyaniline nanomaterial, C_3_N, is able to cause partial denaturation of the globular protein Villin headpiece by disrupting interior hydrogen bonds [77].

### 4.2. Interaction of Noble-Metal-Based Nanomaterials with Proteins

The applications of noble-metal-based nanomaterials, such as Au and Ag, have been extensively investigated due to their remarkable properties [78].

#### 4.2.1. Gold Nanomaterials

Gold nanomaterials have broad applications in nanotechnology due to their high stability and low toxicity. Computational studies have shed light on the molecular mechanisms involved in the interaction between gold nanoparticles and biomolecules, revealing insights into their ability to inhibit the function of proteins associated with various diseases [79]. For instance, researchers have explored the potential inhibitory interaction of gold compounds with thioredoxin reductase 1 (TR1), an enzyme implicated in the growth of tumors as it catalyzes the removal of disulfide bonds in the antioxidant protein thioredoxin [80]. Furthermore, a peptide-coated gold nanocluster demonstrated effective inhibition of TR1 function and the subsequent induction of tumor cell apoptosis in a dose-dependent manner. Molecular docking was used to identify a putative binding region in the vicinity of the catalytic site. Simultaneously, MD simulations were conducted to assess the binding determinants of the peptide-coated gold nanocluster [81]. The results reveal highly charged peptides (+5) employed for functionalizing the metallic nanocluster. Moreover, the driving force behind the binding of the nanocluster to the protein arose from the electrostatic interaction between the positively charged peptides within the nanostructure and the acidic residues near the protein binding site [81].

#### 4.2.2. Silver Nanomaterials

From a computational perspective, substantial progress has been made in understanding the interaction between proteins and silver nanomaterials through a multiscale approach. This approach combines atomistic MD simulations with coarse-grained simulations to investigate the interaction between the ubiquitin protein, which is highly abundant, and citrate-coated silver nanoparticles [82]. Results suggest these interactions arise from electrostatic forces between the nanomaterial and negatively charged protein residues. Additionally, it is significant to mention that the protein structure remains unchanged upon binding, and at high ubiquitin concentrations, an additional protein layer is formed [82].

#### 4.2.3. Palladium Nanomaterials

Palladium-based nanostructures have received less attention than other noble-metal-based nanomaterials. Recent research employing experimental and computational methodologies has unveiled their intriguing antibacterial properties. These properties have been observed in palladium nanocrystals, which exhibit antibacterial efficacy against a spectrum of bacteria, encompassing Gram-negative and Gram-positive strains. It is necessary to highlight that the antibacterial effects depend on the morphology of the nanocrystals [83]. The mechanisms underlying the distinct behaviors of the nanocrystal morphologies have been investigated. Specifically, one morphology exhibits a robust generation of reactive oxygen species, while the other demonstrates superior bacterial membrane penetration. This disparity elucidates the contrasting trends observed in antibacterial efficacy against Gram-negative and Gram-positive bacteria [83].

### 4.3. Interactions of Hydroxyapatite Materials with Proteins

The deposition mechanism of physiologic hydroxyapatite (HA) crystals (mineralization) in collagen-based tissues (bone, dentin, cementum, and calcified cartilage) is a poorly understood complex process [84]. It is generally accepted that during HA formation, both collagen and phosphoproteins (PPs) regulate (promote or inhibit) the growth and proliferation of HA [84].

Intrinsically disordered proteins (IDPs) are proteins whose structures are variable, gaining more folded features when bound to their partners, accounting for ~30% of the human proteome [85]. In vertebrate biomineralization, except for collagen, small leucine-rich proteoglycans (SLRPs), and some enzymes, the majority of proteins associated with HA formation and growth are IDPs, as we recently reviewed [84]. Recently, Tompa employed structural prediction to illustrate that nearly all proteins associated with mineralization in the Swiss protein database were IDPs [86].

The critical interactions between proteins related to biomineralization and HA are not known. However, several MD simulation studies have demonstrated that proteins are generally adsorbed by electrostatic forces of different strengths, depending on the protein structure and surface charge [87,88]. Proteins related to the biomineralization process bind to HA; however, characterizing the adsorption of proteins to solid surfaces is challenging [84]. The energy controlling the absorption of proteins comes from intermolecular forces, such as Coulombic, van der Waals, Lewis acid–base, and hydrophobic interactions. Temperature, pH, ionic strength, protein concentration, and surface polarizability can influence these forces. Together, they control the conformational entropy of the protein structure and the motilities of the protein over solid surfaces [13,89]. Experiments, in silico, suggest that intrinsically disordered peptides absorbed to a surface with a complementary pattern form a well-defined structure (α-helix), indicating that a specific surface can stabilize the structure of an IDP peptide [84]. Furthermore, DPP is a protein found in dentin that is responsible for the formation of HA crystals. DPP mutations can lead to dentinogenesis imperfecta, which results in weaker teeth and a higher risk of tooth decay [90]. Studies have shown that the net charge of DPP peptides affects their structure and affinity to HA surfaces. Peptides that are phosphorylated have a negative charge and show more significant affinity to HA surfaces. It suggests a link between phosphorylation patterns, protein configurations, and HA formation [84,88]. Wendy Shaw’s group demonstrated the effect of a complementary surface pattern on protein structure in vitro [91]. They used a peptide from leucine-rich amelogenin protein (LRAP) to analyze the changes in the secondary structure when this peptide was absorbed into HA and carbonated HA (CAP) [91]. The structure of LRAP peptide absorbed into CAP was consistent with an α-helix, whereas when bound to HA, the structure corresponded more to a random coil [91]. We studied DPP peptides by FTIR spectroscopy and found the phosphorylated peptides in solution in the presence of HA formed α-helical structures and lost their random coil characteristics [84]. Perhaps, through evolution, nature has found that the most efficient strategy for protein–surface interactions is using IDPs to bind to solid surfaces and, in this way, control the nucleation, growth, and morphology of biominerals while still being able to interact with other partners. IDPs do not have a structure to lose on binding and thus can form a more favorable secondary structure on binding [85]. Figure 4 shows a change in structure of two peptides with the same sequence, where one is phosphorylated and the other not. The change in structure is due to the difference in surface charge.

## 5. Basics of QSAR for Biopolymer–Ligand Binding

During the development of drug design, it is crucial to fully understand the conformational barriers involved in the ligand–target recognition process. Such a task can be achieved through several techniques, including X-ray crystallography, NMR spectroscopy, and scanning electron microscopy (SEM) [92,93]. Additionally, computational methods, such as MD simulations, can be used [94]. MD is a computational method commonly used to generate multiple target conformations, allowing significant backbone and amino acid side chain rearrangements [95,96]. It has also been shown that MD yields better data when mixed with docking simulations that allow for the visualization of ligand recognition behavior and for the analysis of free energy values, in comparison with experimental results [95,97].

In the field of drug discovery, many efforts have been made to find ligands that act as either agonists or antagonists. Computational tools, such as molecular modeling (including docking and MD simulations) and QSAR studies, have been useful in designing new drugs [98,99].

The QSAR is the process by which the chemical structure correlates quantitatively with a well-defined process, such as the biological activity (union of a drug with a receptor) or chemical reactivity (affinity of one substance for another to produce a reaction). For example, biological activity can be expressed as the concentration of a substance required to give a specific biological response. In addition, when physicochemical properties or structures are expressed by numbers, we can construct a mathematical relationship, or quantitative structure–activity relationship, between the two. The resulting mathematical expression can then be used to predict the response of other chemical structures [100,101].

The most general mathematical form of QSAR is
Activity=f(physicochemical properties and/or structural properties)

This method calls for the assignment of some parameters to each chemical group, so that by modifying the chemical structure, the contribution of each functional group to the activity of the drug, or the toxicity, in question can be assessed and correlated to how the activity of that substance will vary [102].

The basic assumption of QSAR is that similar molecules will have a similar activity. The underlying problem is how to define a small difference at the molecular level, because each type of activity (for example, chemical reaction capacity, biotransformation capacity, solubility, etc.) could depend on another difference. It has been found that, very often, it is not true that all similar molecules have similar activities [103,104].

The QSAR method requires 3D structures, such as crystallography of known drug molecules, and is called 3D-QSAR. This method uses calculated potentials of a set of small molecules with known activities, which are later superimposed onto experimental data by employing computed potentials such as the Lennard–Jones potential rather than experimental constants. It is focused on the properties of the whole molecule rather than on a simple substituent. When the method examines the steric fields (shape of the molecule) and the electrostatic fields based on the function of applied energy, it is named comparative molecular field analysis (CoMFA) [105].

The dataset of a typical high-throughput screening (HTS) of biologically active compounds can contain over 10,000 compounds. However, these datasets cannot be used directly for modeling purposes due to the presence of duplicates, artifacts, and other issues. Thus, chemical structure curation and standardization is an integral step in QSAR modeling. This process includes the removal of inorganic compounds, structural interconversion, normalization of chemotypes, the treatment of tautomeric forms, and removal of duplicates [106].

A substantial number of molecular descriptors must encode the resulting chemical structures used in QSAR model building. Molecular descriptors are a way in which molecules are transformed into numbers, allowing some mathematical treatment of the chemical information contained in the molecule. Commonly used descriptors are classified in topological, geometrical, thermodynamic, electronic, and constitutional quantities. Generally speaking, the QSAR model is built by using only a few descriptors that are valid for closely related compounds [104].

The data space created is subsequently reduced utilizing an extraction of variables, also called dimensionality reduction. There are many methods that can be applied to achieve this task, such as genetic algorithms or any other learning method such as support vector machines or artificial neural networks [103].

The final part of QSAR model development is model validation, when the true predictive power of the model is established. Obtaining a good-quality QSAR model depends on many factors, such as the quality of input data, the choice of descriptors, and statistical methods for modeling and for validation. Validation is the process by which the reliability and relevance of a procedure are established for a specific purpose [107]. To validate QSAR models, various strategies are adopted: (a) internal validation or cross-validation, (b) external validation by splitting the available dataset into a training set for model development and prediction set for model predictivity checks, (c) blind external validation by application of the model to new external data, and (d) data randomization for verifying the absence of chance correlation between the response and the modeling descriptors [107].

Once the best model is obtained, the next step is to perform a virtual screening on the compound database to find potential drug candidates via molecular similarity approaches and pharmacophore models. This process allows the prediction and selection of the best ligands. The last step is to perform experiments to validate the proposed compounds [104].

Some examples of MD simulations, quantum mechanics simulations, and QSAR modeling applied to molecules of pharmacological interest are described below.

### 5.1. Human Serum Albumin Promiscuity

Proteins are macromolecules consisting of different numbers and sequences of amino acids, which allow them to adopt different 3D structures and possess unique biological functions. Among the most important of these biological functions are biochemical reaction catalysis, cellular signal generation, ligand transport, and structural support. The overall protein structure is flexible under natural conditions. As such, it is possible that regulatory proteins have different binding sites and undergo ligand–binding site conformational changes during the process of ligand binding [108,109]. However, it has been shown that some monomeric proteins are capable of binding to several ligands at different sites. These types of proteins have been termed ligand promiscuous. This behavior is a feature of human serum albumin (HSA), the most abundant plasma protein, and is characterized by its surprising capacity to bind a large variety of biologically active molecules. The reason for the high degree of promiscuity of HSA remains unclear [110]. Deeb et al. [111] performed five-nanosecond MD simulations on HSA to study the conformational features of its primary ligand binding sites (I and II). Additionally, 11 HSA snapshots were extracted every 0.5 ns to explore the binding affinity (Kd) of 94 known HSA binding drugs using a blind docking procedure. MD simulations indicated that there is considerable flexibility for the protein, including the known sites I and II. Structural analyses and docking simulations evidenced movements at HSA sites I and II. The latter enabled the study and analysis of the HSA–ligand interactions of warfarin and ketoprofen (ligands binding to sites I and II, respectively) in greater detail. Those results indicated that the free energy values by docking (Kd observed) depend upon the conformations of both HSA and the ligand. The 94 HSA–ligand binding Kd values, obtained by the docking procedure, were subjected to a QSAR study by multiple regression analysis. The best correlation between the observed and QSAR theoretical (Kd predicted) data was displayed at 2.5 ns. This study provides evidence that HSA binding sites I and II interact specifically with a variety of compounds through conformational adjustments of the protein structure in conjunction with ligand conformational adaptation to these sites. These results serve to explain the high ligand promiscuity of HAS [111].

### 5.2. Antimicrobial Peptides

Over the last decade, several disease-causing microbes have become resistant to antibiotic drug therapy and created a significant public health problem. Some of these organisms resistant to all approved antibiotics are related to diseases such as tuberculosis, pneumonia, gonorrhea, septicemia, wound infections, and biofilms, and, to date, they can only be treated with experimental and potentially toxic drugs. Therefore, there is an urgent need to develop new antimicrobial drugs to surmount the increasing resistance of pathogens to existing antibiotics [112]. Naturally occurring antibiotics are the right place to start. Molecules with such potential are the antimicrobial peptides (AMPs) found in insects, animals, and plants since they have functioned as an essential component of the natural innate immune system of living organisms against pathogens, and most importantly, the microorganisms have not developed resistance. Moreover, many AMPs act mainly on bacterial membranes, unlike most antibiotics, which usually target specific proteins [113,114].

Velasco-Bolom et al. [113] performed 12 μs all-atom MD simulation of the AMPs Pandinin 2 (Pin2) and Pin2GVG to explore their adsorption mechanism and the role of their constituent amino acid residues when interacting with pure POPC and pure POPG membrane bilayers. Pin2 is an alpha-helical polycationic peptide, identified and characterized from the venom of the African scorpion Pandinus imperator with high antimicrobial activity against Gram-positive bacteria and less active against Gram-negative bacteria; however, it has demonstrated strong hemolytic activity against sheep red blood cells. In the chemically synthesized Pin2GVG analog, the GVG motif grants it low hemolytic activity while keeping its antimicrobial activity [81,113]. In Figure 5, the results of MD simulations show that Pin2 and Pin2GVG affect the surfaces of membranes differently depending on their composition.

Starting from an α-helical conformation, both AMPs are attracted at different rates to the POPC and POPG bilayer surfaces due to the electrostatic interaction between the positively charged amino acid residues and the charged moieties of the membranes. Since POPG is an anionic membrane, the PAM adhesion is stronger to the POPG membrane than to the POPC membrane, is stabilized more rapidly. This study reveals that, before the insertion begins, Pin2 and Pin2GVG remained partially folded in the POPC surface during the first 300 and 600 ns, respectively, while they were mostly unfolded in the POPG surface during most of the simulation time. The unfolded structures provide for many intermolecular hydrogen bonds and stronger electrostatic interactions with the POPG surface. The results show that the aromatic residues at the N-terminus initiate the insertion process in both POPC and POPG bilayers. As for Pin2GVG in POPC, the C-terminus residues seem to initiate the insertion process, while, in POPG, this process seems to be slowed down due to a strong electrostatic attraction. The membrane conformational effects upon PAM binding are measured in terms of the area per lipid and the contact surface area. Several replicas of the systems lead to the same observations [113,114]. Figure 6 shows the pore formed after 50 ns of coarse-grained MD simulations of a Pin2 oligomer. The pore diameter is narrower in the center and shows an affinity for Cl^−^ ions.

### 5.3. Human Vasopressin V1a Receptor

Arginine–vasopressin (AVP) is a nonapeptide hormone secreted mainly from the posterior pituitary gland. AVP exerts multiple biological actions both as a hormone and as a neurotransmitter. AVP acts on three different vasopressin receptors (VPRs) that belong to the GPCR group: the V1a, V1b, and V2 receptors. V1aR is mainly expressed on platelets, the liver, and vascular smooth muscle, where it plays an essential role in platelet aggregation, glycogenolysis, and vascular contraction. Several peptide analogs of AVP have been used as potent VPR antagonists, despite their poor oral bioavailability and reduced therapeutic effect. Currently, non-peptide VPR antagonists (vaptans) are being investigated for clinical use [115,116].

Contreras-Romo et al. [115] studied a set of 134 V1aR antagonists using molecular modeling and QSAR studies to identify the ligand recognition properties that explain their pharmacological effects.

The docking results suggest that the ligand recognition of V1aR is mediated by p–p and p–cation interactions with F207, W211, and R214, which become critical residues in the binding pocket of this receptor. Residues such as V217, I310, and Q311 engage in hydrogen bonds and hydrophobic interactions, suggesting an important participation in ligand recognition. Hydrogen bond interactions with the N atom of side chain of R214 suggest that a heterocyclic ring in the ligand structure is needed for a high affinity to V1aR. SR121463 is not benzodiazepine derivative, but its affinity for V2R is approximately 100 times greater than for V1aR. Similarly, SSR149415 also lacks the heterocyclic ring, but it has high V1bR antagonistic activity. Thus, the data coupled with the QSAR studies suggest that heteroatoms, in azepines such as N and O, are essential structural components for recognition on V1aR [115].

## 6. Conclusions

Several issues hinder the understanding of how living beings or biomolecules interact with biomaterials in biomedical engineering, how we can improve the biocompatibility of biomaterials of different natures, and, equally important, how we can understand the responses of materials when they are in contact with a biological environment. Therefore, it is of great importance to predict the intimate interactions between biomaterials and living systems. This is fundamental to understanding water organization around biomaterials, how these waters layers can affect the absorption of core proteins, and their folding. A developed surface that allows rapid absorption of proteins in their native configuration and where cell adhesion sites are exposed will allow a better cell differentiation, proliferation, and migration, which are essential for tissue regeneration. The generation of models helps us generate data for a greater prediction of the systems.

Computational techniques such as molecular dynamics simulations have been used to gain a better understanding of these interactions, biomolecules, and biomaterial surfaces, which could affect the functionality of the biomaterials in biological systems. An ideal modeling should explain the mechanisms by which a biomaterial can induce biological or adverse effects. These models also should predict the likelihood of a given effect happening and its magnitude, and then infer the properties (biological effects, physical and mechanical properties) of this system and other similar systems not yet tested [76]. Bioengineers must be able to answer these questions on the interface between the body and the surrounding environment at the nano, micro, and meso scales.

In summary, molecular modeling plays a pivotal role in advancing our understanding of the interactions at polymer–water interfaces. While significant strides have been made in this field, the journey towards a complete comprehension of the roles of moisture, morphology, and mechanics is ongoing. Nevertheless, computational modeling remains an invaluable tool that continually evolves and pushes the boundaries of knowledge in biopolymers with biomedical applications.

## Figures and Tables

**Figure 1 polymers-16-01864-f001:**
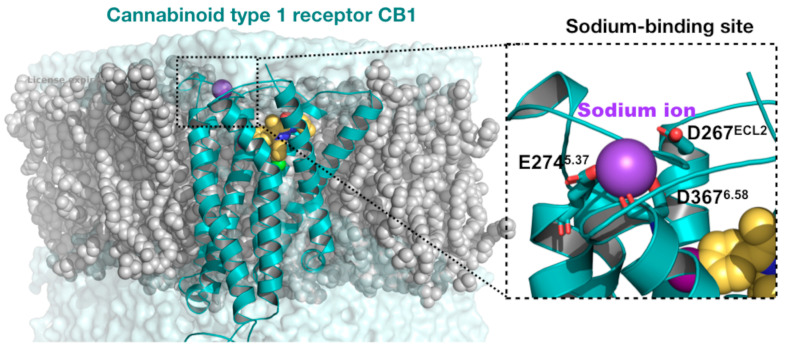
All-atom MD simulation of the CB1 receptor with a new extracellular sodium-binding site identified.

**Figure 2 polymers-16-01864-f002:**
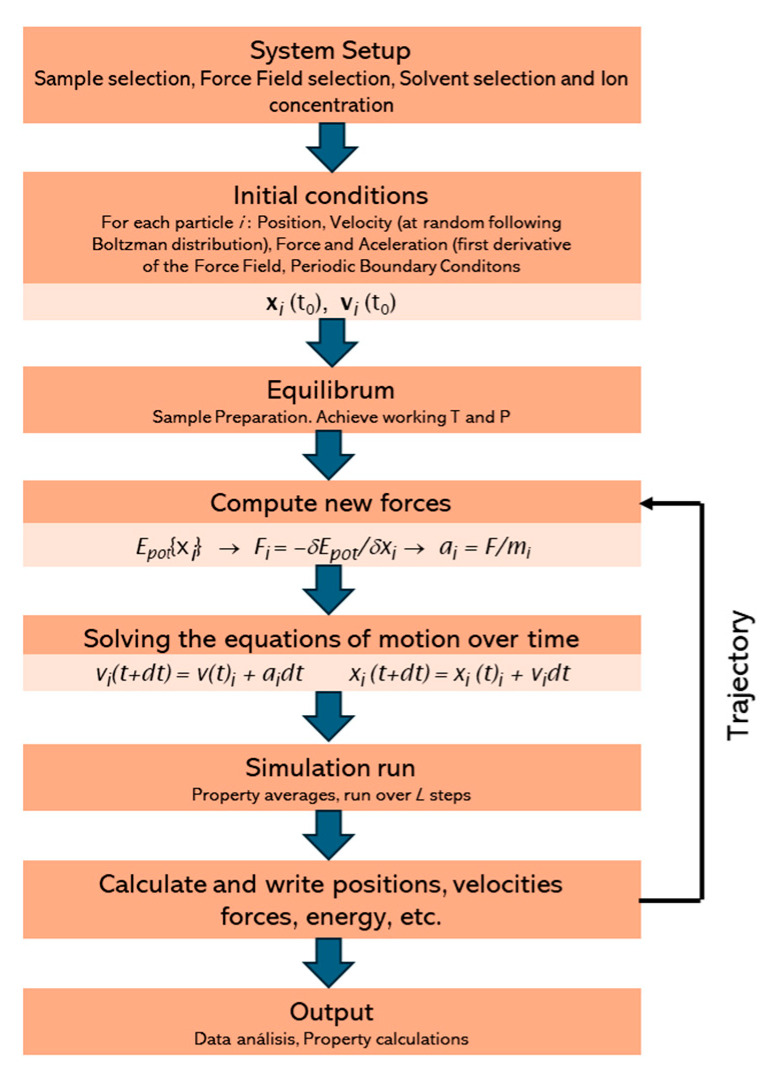
Computational workflow of MD simulations.

**Figure 3 polymers-16-01864-f003:**
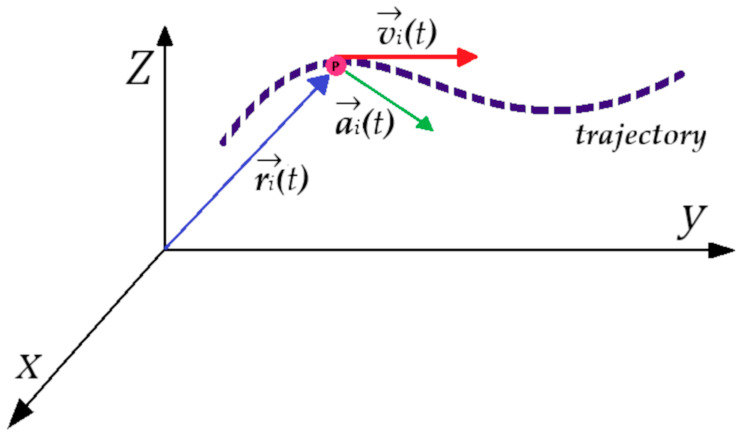
During MD simulation, the trajectory of a particle is represented by a dotted line. The position vector is denoted with a blue arrow, the velocity vector with a red arrow, and the acceleration vector at time *t* with a green arrow.

**Figure 4 polymers-16-01864-f004:**
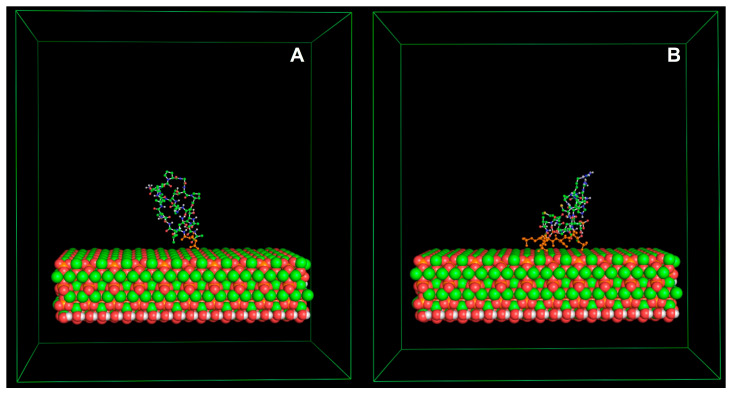
The HA surface is displayed as spheres, whereas the peptides with the same amino acid sequence are represented as balls and sticks. Orange amino acids represent negatively charged and phosphorylated amino acids in the peptides. (**A**) Non-phosphorylated peptide. (**B**) Phosphorylated peptide. A change in structure is observed due to the different surface charges of the peptides.

**Figure 5 polymers-16-01864-f005:**
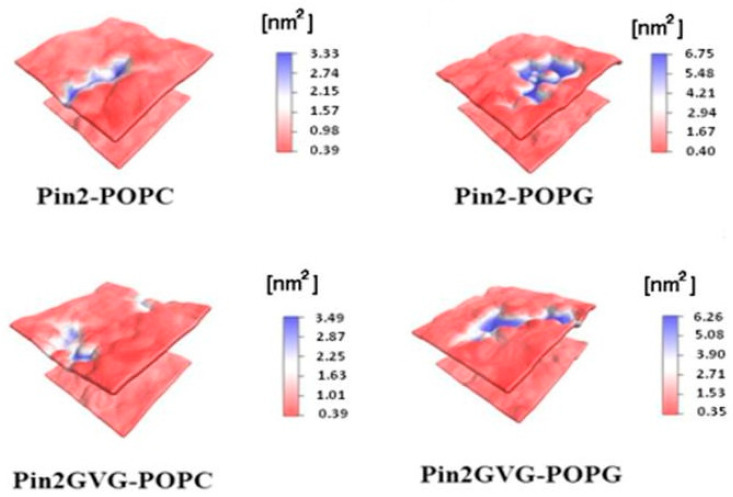
The results of the MD simulations show that Pin2 and Pin2GVG affect the surface of the membranes differently depending on their composition.

**Figure 6 polymers-16-01864-f006:**
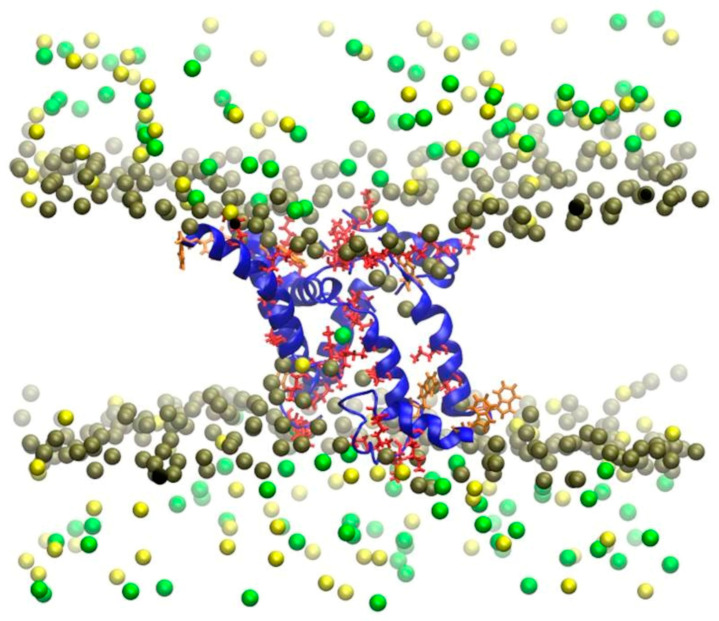
Pore formed after 50 ns of coarse-grained MD simulations of a Pin2 oligomer. The pore diameter is narrowest in the center and has an affinity for Cl^−^ ions. Water molecules and phospholipid chains have been omitted for clarity. The gray spheres are the phosphate headgroups, the green spheres are Cl^−^ ions, and the yellow spheres are Na^+^ ions.

**Table 1 polymers-16-01864-t001:** List of the programs available for MD simulations along with their corresponding links.

List of MD Programs	Link
MDWEB	https://mmb.irbbarcelona.org/MDWeb/
WEBGRO	https://simlab.uams.edu/
VISUAL DYNAMICS	https://visualdynamics.fiocruz.br/login
GROMACS	https://www.gromacs.org/
NAMD	https://www.ks.uiuc.edu/Research/namd/
CHARMM	https://www.charmm.org/
LAMMPS	https://www.lammps.org/
AMBER	https://ambermd.org/
MDN	http://mdn.cheme.columbia.edu
TINKER	https://dasher.wustl.edu/tinker/
OPENMD	https://openmd.org/

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
