# Peer review of "Molecular Dynamic Simulations for Biopolymers with Biomedical Applications"

_polymers, 2024, doi:10.3390/polym16131864_

Round 1

Reviewer 1 Report

Comments and Suggestions for Authors

I think that this work can be considered to publish after clearing the following comments.

1.When mentioning the three key research areas of CM in biopolymer biomedical engineering (Computer-aided design, Finite Element Analysis, and Molecular Dynamics simulations), it is suggested to provide brief descriptions or examples for each area to help readers better understand their importance and applications.

2.When mentioning that MD simulations can solve Newton's equations of motion and generate spatial trajectories for each atom, it is suggested to provide some specific examples or illustrations to make the description more vivid and easier to understand.

3.When providing a comprehensive overview of the diverse applications of MD simulations on biopolymers, it is recommended to categorize them according to some logic or theme, such as the type of biopolymer or the application area of MD simulations. This will make the review more organized.

4.When emphasizing the flexibility, robustness, and synergistic relationship between in silico and experimental techniques, it is suggested to provide some specific examples or case studies to support these points. This will enhance the persuasiveness of the paper.

5.Check and ensure that all technical terms are used accurately and appropriately.

Author Response

Dear Reviewer 1,
We sincerely appreciate your insightful comments, which have proven effective in uncovering previously overlooked errors. Embracing your feedback, we have implemented changes to the enclosed file. These additions were highlighted in yellow, and deletions indicated with strikethroughs and highlighted. Furthermore, the Reviewer's remarks have been integrated in red and highlighted to enhance the clarity of our changes. Your contributions are deeply appreciated. Thank you kindly, Eduardo. 

We corrected the following lines: 26 to 37, 156 to 172, 203 to 265, 304 to 323, and 566 to 575. We also added Figure 2.

Reviewer 2 Report

Comments and Suggestions for Authors

The article comprehensively describes the issue of "Molecular Dynamic Simulations for Biopolymers with Biomedical Applications". It should be treated as a review article, and in this form, in my opinion, it meets the requirements of the journal. The literature review contains a lot of information proving the authors' deep knowledge and good preparation of the review material. The cited literature is extensive and relatively modern. No major editing mistakes.

Author Response

Dear Reviewer,
We sincerely appreciate your insightful comments, which have proven effective in uncovering previously overlooked errors. Embracing your feedback, we have implemented changes to the enclosed file. These additions were highlighted in yellow, and deletions indicated with strikethroughs and highlighted. Furthermore, the Reviewer's remarks have been integrated in red and highlighted to enhance the clarity of our changes. Your contributions are deeply appreciated. Thank you kindly, Eduardo. 

Reviewer 3 Report

Comments and Suggestions for Authors

The manuscript is a review of the application of molecular dynamics (MD) methods to simulate the behavior of biopolymer macromolecules in their interaction with various microobjects. This makes it possible to evaluate a particular biopolymer for certain biomedical applications. The topic is relevant, important, and the manuscript could be recommended for publication. However, some additional work is needed.

It appears that parts of the manuscript written by different authors were not properly edited after putting together. This manifests itself, in particular, in multiple definitions of the same abbreviation. The abbreviation “MD” is defined twice in the Abstract (lines 28 and 34) and multiple times in the text below (lines 92, 127, 208, 214, 263 and 568), “CNTs” is in lines 417 and 421. The abbreviation “NMR” is defined on line 567, and is used above, on line 58. “Hydroxyapatite (HA)” is defined on line 524, after its double use somewhere above. “QSAR” appears on lines 563 and 576, being defined on line 578 and redefined on line 667.

On line 261 the authors mention “software packages CHARM, AMBER, and GROMOS”, but where is a list of programs, free and proprietary, used for MD? The authors of numerous papers couldn’t re-program MD algorithms every time! A list of programs available on the Internet, with recommendations, would be very useful for the reader planning to engage in computer modeling of biopolymers.

Mathematical formulae are typed carelessly, with typos, and equation numbers are not aligned to the right. In Eq. (2.1), the quantity being differentiated has become a subscript and is no longer bold, even though it is a vector. The letter “U” in lines 164 and 167 looks different. One of the subscripts (j) is missing in Eq. (2.3). On line 186, scalar radii should not be bold. Lines 194 and 195: “the kinetic energy and temperature of the system are determined by velocity vi(t)”—no, these quantities are determined not by the velocity of the ith particle alone, but by the velocities (moments) of all particles in the system. The time t does not need to be typed in bold (lines 194, 195, 199, and 205).

The references to Figs 1–3 are sloppy, just “Figure N.” not in parentheses (lines 558, 697 and 718). The figure captions look strange, being full phrases with a verb in the personal form (as in popular science articles) instead of just titles, “Figure N” is duplicated (lines 560 and 720), there are no references to the sources of the figures.

There are minor typos as well. “[72] [102]” on line 697; in lines 700, 706 and 716, instead of AMPs there are PAMs (these are polyacrylamides, by the way). In Subsection 4.1.1, subscripts are not formalized (lines 393, 398, 401, 403, 410, 411, etc.). Line 463 does not have a superscript (sp2). Line 140: “was first introduced by Car and Parinello in 1985” (no citation, the reader has to search for the relevant paper in the extensive bibliography). Everything taken together spoils the impression of the manuscript and needs to be eliminated.

Author Response

Dear Reviewer 1,
We sincerely appreciate your insightful comments, which have proven effective in uncovering previously overlooked errors. Embracing your feedback, we have implemented changes to the enclosed file. These additions were highlighted in yellow, and deletions indicated with strikethroughs and highlighted. Furthermore, the Reviewer's remarks have been integrated in red and highlighted to enhance the clarity of our changes. Your contributions are deeply appreciated. Thank you kindly, Eduardo. 

We corrected the following lines: 38 to 46, 79, 117, 153, 186, 210 to 273, 299, 343, 427 to 433, 441, 479, 484, 487, 489, 497, 503, 507, 553, 621, 623, 662, 666 to 668, 677, 679, 768, 797 to 799, 815 to 817, and 893 to 894. We also added Table 1 and Figure 2 .
